# Influence of Secondary Reinforcement on Behaviour of Corbels with Various Types of High-Performance Fiber-Reinforced Cementitious Composites

**DOI:** 10.3390/ma12244159

**Published:** 2019-12-11

**Authors:** Nasuha Md Zin, Amin Al-Fakih, Ehsan Nikbakht, Wee Teo, Mahmoud Anwar Gad

**Affiliations:** 1Civil and Environmental Engineering Department, Universiti Teknologi Petronas, Bandar Seri Iskandar 32610, Malaysia; nasuhamdzin@gmail.com; 2Infrastructure and Society (EGIS), Heriot Watt University Malaysia, Putrajaya 62200, Malaysia; t.wee@hw.ac.uk; 3Civil Engineering Department, Al-Azhar University in Cairo, Cairo 11751, Egypt; mahmoud.anwar@azhar.edu.eg

**Keywords:** corbel, ECC, hybrid fiber-reinforced concrete (HyFRC), steel fibers, PVA fibers, primary reinforcement, secondary reinforcement

## Abstract

An experimental study is conducted to determine the influence of secondary reinforcement on the behaviour of corbels fabricated with three different types of high-performance fiber-reinforced cementitious composites, including engineered cementitious concrete (ECC); high-performance steel fiber-reinforced composite (HPSFRC); and hybrid fiber-reinforced composite (HyFRC). Two shear span-to-depth ratios (a/d = 0.75 and 1.0) are explored. The mechanical properties of the composites in terms of tensile, compressive, and flexural strengths are investigated. Next, the structural behaviour of the high-performance cementitious composite corbels in terms of ultimate load capacity, ductility, and failure modes under the three-point bending test are investigated. The secondary reinforcement is proven to significantly affect stiffness and ultimately load capacity of all three high-performance composite corbels with an aspect ratio of 0.75. However, the secondary reinforcement was more impactful for the HPSFRC corbels, with 51% increase of ultimate strength. Moreover, in terms of damage, fewer cracks occurred in ECC corbels. HPSFRC corbels displayed the highest level of ductility and deformation capacity compared to the other specimens. The results were comparatively analyzed against the predicted results using truss and plastic truss models which provided relatively reliable shear strength.

## 1. Introduction

A corbel is generally described as a short member with shear span-to-depth ratio lower than the unity that cantilevers out from a column or a wall [1]. Brackets and corbels may be described as short cantilevers that work as a connector between columns and beams to support the heavy load of cranes and precast beams. Internal primary and secondary (closed stirrups) reinforcements are incorporated into the design of the corbels to prevent brittle failure under excessive horizontal and vertical loads. Several studies have been performed, both experimental and analytical, on the main parameters governing the structural behaviour of corbels, such as reinforcement ratio, shape and geometry, and concrete strength [2,3,4]. When the ratio of primary reinforcement is low, it enables the members to achieve higher ductility because of yielding of reinforcement prior to the concrete. It has been shown that the addition of secondary reinforcement generally results in enhancing the strength and deformation capacity of corbels and leads to a less catastrophic failure [5,6,7,8]. However, in members with congested reinforcement in a small area, other problems might occur, such as honeycomb, voids, and inadequate bonding between the surface of the reinforcement and the concrete, which cause premature failure of the structures. Moreover, several researchers reported that incorporating normal concrete and stirrups as secondary reinforcement is associated with a significant loss in the strength of corbels after reaching the peak load [9].

The behaviour of high-strength concrete corbels has been investigated in the presence or absence of steel fiber up to a concrete compressive strength of 132 MPa [10,11,12,13]. Despite the development of novel theoretical models, empirical methods, and common detailing practices remain the typically used approaches since there is no consensus on the most effective design model.

The nonlinear stress behaviour of the short member is affected by the shear deformation in the elastic range, and consequently the shear strength of the section becomes an important parameter for design consideration [2,14]. The principal failure mode for corbels reinforced with principal and secondary reinforcements (stirrups) is referred to as beam-shear failure, which is characterized by the cleaving or splitting of one or several diagonal cracks accompanied by shear failure in the compressed region of the strut [15]. Moreover, corbels exhibit many distinctive modes of failure, which include yielding of the tension tie; failure of the end anchorages of the tension tie, either under the load point or in the column; failure of the compression strut by crushing or shearing; and localized failures under the bearing plate [16].

For uniaxial compressive behaviour, several experimental works and numerical models have been conducted to investigate the impact of incorporating fiber into normal and high-strength concrete [17]. The ductility of the stress–strain curves was shown to increase concomitantly with an increase in fiber volume fraction and/or fiber aspect ratio [18]. The fibrous concrete attained a marginal increase in compressive strength of approximately 9% as compared to non-fibrous concrete.

To limit the defects typically observed in the conventional concrete, as well as decrease the quantity of internal reinforcements, high-performance fiber-reinforced cementitious composite (HPFRCC) was introduced. Throughout the last two decades, various HPFRCCs using different fibers have been investigated. HPFRCCs generally retain a fiber volume fraction that is lower than 2% and display a pseudo-strain-hardening behaviour under tensional conditions, and improved tolerance for damage [19,20]. These enhanced mechanical properties of HPFRCC are mostly attributable to the increased inter-particle bonding; the utilization of diverse types of fibers, such as steel and polyvinyl alcohol (PVA), and implementation of low water–cement ratio in the mixture design.

Moreover, hybridization of different fiber types at low volume can significantly improve the performance of concrete corbels due to the combined effect of micro- and macrofibres [21]. This will enhance both mechanical performance and durability. The exceptional properties of these materials make them a practical option for advancing the structural performance of concrete corbels under severe loadings.

To promote the practical and economic benefits of using fibers in concrete corbels, specific aspects associated with their behaviour and design have to be well researched. A vital aspect is to determine the behaviour of corbels under different loads, and their controlling parameters [21].

To date, several studies investigated the structural behaviour of high-performance self-compacting concrete (HPSCC) structural members; however, there is limited research conducted on the influence of closed stirrups (secondary reinforcement) on various types of HPSCC corbels including hybrid fiber-reinforced high-performance cementitious composites. In this study, the structural behaviour of corbels with engineered cementitious concrete (ECC), steel fiber-reinforced concrete (SFRC) and hybrid fiber-reinforced composite (HyFRC) in different shear span-to-depth ratios of 0.75 and 1.0 are investigated. The influence of secondary reinforcement on force–displacement behaviour, displacement ductility, and failure modes of corbels made of HyFRC, ECC, and SFRC is examined and compared. For this purpose, corbels with different shear span-to-depth ratio (a/d) are explored. Moreover, the validity of existing analytical models is examined and comparatively analyzed with the results from the experiments on the high-performance cementitious composite corbels investigated in this study.

## 2. Experimental Program

### 2.1. Specimens and Testing Setup

The dimensions and labelling of the specimens are shown in Figure 1 and Figure 2, and Table 1. In this figure, there are two series of “P” and “S” corbels. “P” represents the corbels with only primary reinforcements, whereas “S” represents the corbels with primary and secondary reinforcements, e.g., corbel P 0.75 indicates the corbels without secondary reinforcement and aspect ratio (a/d) of 0.75.

As shown in Figure 1, the heights of the corbels are 160 mm and 200 mm for the specimens with aspect ratios of 0.75 and 1.0, respectively, where the corbels are placed on the steel roller supports at the distance of 150 mm and 160 mm from the face of columns. The width of all corbels is 120 mm.

In this study, a total of 12 corbels are tested. The main variables of this study include the presence of secondary reinforcement (stirrup); and shear span-to-depth ratio (a/d). In this study, two aspect ratios of 1.0 and 0.75 were selected. Using the test setup displayed in Figure 3, the samples were subjected to three-point loading test at a constant rate of 1.5 mm/min. For each load increment, a reading was recorded until the occurrence of failure. The corbels are tested in an inverted position using a UTM 500 kN capacity testing machine. The vertical load is applied to the top of the column by means of a self-supporting loading frame of the universal hydraulic testing machine where load cell was attached to the hydraulic jack to measure the applied load. The vertical displacement was monitored by two linear variable differential transformers (LVDTs) placed at the left and right of both sides of the corbel.

### 2.2. Mechanical Properties

The ingredients/components of the mixture designs for the HyFRC, ECC, and SFRC used in this study are shown in Table 2. The admixtures for the three kinds of HPCCs were chosen with the intent of maintaining self-compacting properties, pseudo tensile strain hardening, and attaining similar splitting tensile strengths. Characteristic compressive strengths for HPCC materials that are consistent with this classification vary from 35 MPa [22] to 80 MPa [23] with tensile strengths that range from 2.0 MPa [24] to 7 MPa [23].

The total fiber volume fraction of 1.6% is used in the mixture designs of all the cementitious composites. The admixtures are selected so that the compressive strengths of all cementitious composites are in the range of 50–65 MPa. In the HyFRC, fiber hybridization consisted of polyvinyl alcohol (PVA) fibers and steel fibers. The properties of reinforced bars used in the investigation is shown in Table 3. Additionally, the properties of steel and PVA fibers used in this study are shown in Table 4 and Figure 4.

The flexural, tensile, and compressive strengths of all HPCCs were obtained. Flexural strength, referred to as modulus of rupture, is the capacity for material to resist deformation when subjected to loading. The flexural strength test was performed based on ASTM C293-16 [25] with center-point loading for all specimens using Universal Testing Machine, INSTRON. The testing procedure was maintained at a speed of 2 mm/min. Three prism samples that are sized 100 mm × 100 mm × 600 mm were prepared for each type of concrete, which were then tested after curing for 7 and 28 days. The prism sample was placed flat over two points of contact. Afterwards, a force was applied to the center upper part of the sample until failure was attained.

The indirect tensile strength is the ability of the concrete to withstand in a pulling force. The test was performed according to ASTM C496 [26]. After 7 and 28 days of curing, the tensile strength of three concrete cylinders (100 mm × 200 mm) from each mixture were tested using the indirect tensile test machine as shown in Figure 5.

The compressive strength test was carried out following the procedures of ASTM C39-12 [27]. Three concrete cubes (100 mm × 100 mm × 100 mm) were prepared and tested using the compression testing machine after curing for 7 and 28 days. The load was progressively applied without shock at the rate of 140 kg/cm^2^/min until failure was attained.

The result of flexural, tensile, and compressive strength of all specimens are shown and compared in Table 5. As shown in the table, all samples with different types of high-performance cementitious composites achieved similar tensile strengths. Moreover, a sample of failures of the specimens is shown in Figure 6. As can be seen from the figure, none of the specimens exhibited a brittle failure. In terms of flexural mechanical properties, the specimens with steel fiber failed with a wide crack opening; however, due to the bridging of the cracks by the steel fibers, it did not break into two parts. In contrast, the ECC specimens with PVA displayed smaller width of crack openings. According to the results shown in Table 5, the flexural strength of the specimens was between 11 and 15 MPa. ECC displayed slightly lower flexural strength compared to SFRC and HyFRC, i.e., 10.58 MPa flexural strength of ECC versus 15.02 MPa and 12.45 MPa flexural strengths of steel fibre-reinforced composites, SFRC and HyFRC, respectively. Moreover, as stated in Table 5, the compressive strength of the specimen with only steel fiber (SFRC) was 65 MPa; whereas the specimens with PVA (HyFRC and ECC) obtained 58 MPa and 50 MPa, respectively.

Furthermore, in this study, a slump flow test is conducted to measure the workability of the specimens based on the slump test procedure for self-compacting concrete described in EN 1235-2. This test is conducted by using Abrams cone with a base diameter of 200 mm, top diameter of 100 mm, and height of 300 mm. The flowability is measured by the average of maximum diameter of the concrete spread ($D_{\max})$ and its corresponding perpendicular diameter ($D_{\mathrm{perp}})$. Moreover, the flow rate of the samples is measured by the time needed for the flow to reach 500 mm diameter (T500). As stated in Table 6, the flowability of the samples ($D_{\mathrm{avg}})$ were ranged from 693 mm to 771 mm which is within the flowabilty of self-compacting concrete [28]. Among the specimens, the ECC sample showed the highest flowability of 771 mm, followed by HyFRC and SFRC with 721 mm and 693 mm, respectively. A sample of the slump flow test conducted is shown in Figure 7.

## 3. Results and Discussion

Figure 8 represents the results of the load-displacement responses of the ECC, HyFRC, and SFRC corbels with and without secondary reinforcement. As the results indicate, except for the S 0.75 specimen of the SFRC corbel, all other corbels exhibited a high level of ductility and strength. Moreover, unlike the conventional concrete corbels reported in the literature [9], there is no considerable loss in the strength of all HPFRCC corbels after the peak load. As can be observed from the figure, the stiffness and strength of the corbels are varied. In general, the corbels with lower shear span to depth ratio (a/d) exhibited higher stiffness compared to the corresponding specimens with lower a/d. Additionally, the corbels with secondary reinforcement exhibit slightly higher stiffness and more ductile behaviour than the specimens without secondary reinforcement. Furthermore, there is a slight increase in the slope of load-displacement responses approximately after 1 to 2 mm of displacement in the graphs, which occurs at the early stage when cracks initiate at the tension zones. This increment is due to the bridging effect of fibers across the cracks, which results in tensile strain hardening and consequently a slight increase of stiffness in the cracked zone in concrete.

As can be seen from the figure, specimens S 0.75 in both SFRC and ECC exhibited approximately similar load capacities of 132 kN and 128 kN, respectively. However, this specimen for HyFRC showed considerably lower strength of 108 kN capacity. This lower strength can be associated with the weaker bonding strength between the particles of HyFRC due to less amount of fly ash in its admixture, as shown in Table 2. However, this specimen of SFRC exhibited brittle behaviour after reaching the peak load, whereas the other two corbels of HyFRC and ECC show greater ductility. Therefore, when shear is more dominant, i.e., a/d = 0.75 and secondary reinforcement exists (P 0.75), the ECC corbel shows superior performance with a high level of strength without compromising ductility, as compared to SFRC and HyFRC corbel specimens. However, for the specimen without secondary reinforcement, P 0.75, all corbels showed approximately similar strength. On the other hand, when the flexural is dominant (a/d = 1.0) and when there is secondary reinforcement, i.e., S 1.0, the SFRC corbels showed the lowest stiffness and ultimate strength compared to that of corresponding ECC and HyFRC corbels, i.e., the SFRC corbel exhibited 61 kN ultimate strength versus 78 kN and 75 kN strengths of ECC and HyFRC corbels, respectively. However, when there is no secondary reinforcement, similarly to the a/d = 0.75 specimens, P 1.0, the ECC, HyFRC, and SFRC corbels showed approximately similar behaviour and ultimate strengths.

Figure 9 compares the strength increment of HPFRCC corbels due to the presence of secondary reinforcement. As can be observed from the figure, increases of the strengths for the corbels with a/d = 0.75 are more significant. Furthermore, as can be seen from the figure, the SFRC corbel is the most affected among the other cementitious composite corbels, with 51% increment in ultimate strength, followed by the ECC corbel with 45% and lastly the HyFRC with 29% increment. However, for the a/d of 1.0, this order is reversed, where the most impactful corbel is HyFRC with 39%, then ECC corbel with 25% and lastly SFRC with only 11% increment.

Moreover, the displacement ductility ratio in this study is calculated from the method adopted by [29,30], as shown in Figure 10. The displacement ductility of all HPFRCC corbels is compared in Figure 11 and Table 7. Comparing the S series with the P series, it is obvious that the presence of secondary reinforcement slightly increases the displacement ductility of corbels. However, it has a negligible effect on ductility of ECC corbels. Moreover, as shown in Figure 11, ECC has the lowest ductility among other HPFRCC corbels. The ductility of all ECC corbels for both a/d of 0.75 and 1.0 is approximately constant between the range of 2 to 2.5. In contrast, the SFRC corbels showed the highest level of displacement ductility, followed by HyFRC corbels.

The crack patterns and failure modes of all ECC, HyFRC, and SFRC corbels are depicted in Figure 12. As observed, fewer cracks and only minor damage occurred in ECC and HyFRC corbels, whereas the SFRC suffered the highest level of damage and cracks. In addition, as shown in Figure 13, shear cracks emerge in the upper segment of specimen S 1.0 and P 0.75 in SFRC corbels, although this was not identified in the other samples in ECC and HyFRC corbels. The appearance of cracks in S 1.0 can be attributed to the high level of displacement ductility in the sample, which induces significant amounts of concrete cracks at higher level of displacements. Moreover, the P 0.75 sample in SFRC displayed a high level of stiffness and strength prior to the failure. Consequently, excessive stress induced to the concrete and shear cracks are propagated to the compression zone of the specimen.

In general, flexural cracks first emerge in all corbels at the region close to the junction between the column face and the tension face of the corbel. Afterwards, the cracks propagate and expand as the load increases. The cracks for corbels with secondary reinforcement started at the re-entrant corner, and although the crack is propagated laterally along the column–corbel interface, a subsequent crack is initiated at the interior edge of the bearing plate. Conversely, in the absence of secondary reinforcement, the crack is initiated at the inner boundary of the bearing plate, and a second crack originates at the junction of the column. As shown in this study, provision of secondary reinforcement reduces crack width and amount of cracks for SFRC corbels when the shear is more dominant (a/d = 0.75), which is similar to the behaviour of high strength corbels and ultra-high performance concrete (UHPC) corbels with steel fibre [8,11,13]. However, the specimens with a/d of 1.0, where flexure is more dominant, the provision of secondary reinforcement causes more numerous cracks as shown in Figure 12. Moreover, as displayed in the figure, compared to SFRC corbels, fewer and finer cracks induced to ECC and HyFRC corbels, which make them superior in terms of durability, service life, and sustainability [8].

## 4. Analytical Analysis

The mathematical model to predict the shear strength of fiber-reinforced cementitious composite corbels is presented in this section. As shown in Section 3, the difference in ultimate shear strengths of the corresponding HPCC corbels with steel and PVA fibers for SFRC, ECC, and HyFRC slightly varied but were not significantly different especially when there are no secondary reinforcements. Hence, in this section, the ultimate strengths results discussed earlier in this study are compared with the Fattuhi model [7] and Foster model [12] in order to investigate the validity of existing formulas to predict the ultimate strength of self-compacting HPCC corbels with steel and PVA fibres.

### 4.1. Fattuhi Model

Fibers can be considered as minute reinforcements dispersed in the concrete structure. The Fattuhi model was developed based on the experimental studies on the shear reinforcement of concrete corbels with steel fibers. The Fattuhi model is utilized for the prediction of the shear strength of fiber-reinforced concrete corbels. The reinforcement of concrete with fibers can preserve tensile stresses following the cracking. The tension preserved by the reinforcement is denoted by a resultant tensile force $T_{f}$, which is presumed to act similar to the core steel or stirrup forces (illustrated in Figure 14). In predicting the shear capacity of the fiber reinforced corbels based on the equations developed by Fattuhi [7], certain parameters of the fiber must be taken into consideration, such as material properties, aspect ratio, fiber composition, and shape. Three formulas have been developed and suggested by Fattuhi and were applied to find the ultimate shear strength of fiber-reinforced cementitious composite corbels.
(1)$$l\sin\beta =\frac{f_{ys}A_{s}+f_{yi}A_{si}+k_{o}f_{ct}bh}{0.85f_{c}^{\prime }b+k_{o}f_{ct}b}$$
where β = the angle between the compressed concrete strut and the vertical direction (Figure 14b), l = the width of the compression force ($C_{c}$) in the inclined strut of the corbel, mm, $f_{ys}$ = yield strength of the main tension reinforcement, MPa, $A_{s}$ = tension reinforcement area, mm^2^, $f_{yi}$ = distribution reinforcement yield strength, MPa, $A_{si}$ = distribution reinforcement area, mm^2^, $f_{ct}$ = split tensile strength, MPa, $k_{o}$ = fibrous concrete contribution in tension, where $k_{o}=9.519/f_{c}^{\prime^{0.957}}$.
(2)$$0.425f_{c}^{\prime }b{\left(l\sin\beta x\right)}^{2}\cot^{2}\beta +0.85f_{c}^{\prime }ab(l\sin\beta )\cot\beta -f_{y}A_{s}\left[d-\frac{l\sin\beta }{2}\right]-\;f_{yi}A_{si}\left[d_{i}-\frac{l\sin\beta }{2}\right]-0.5k_{o}f_{ct}bh[h-(l\sin\beta )]=0.$$

By applying the properties of the corbel samples, the value of (l sin β) can be calculated from Equation (1). Afterwards, cot β can be derived from Equation (2). The shear capacity can be calculated from Equation (3).
(3)$$V_{u}=\frac{f_{y}A_{s}\left[d-\frac{l\sin\beta }{2}\right]+f_{y_{i}}A_{si}\left[d_{i}-\frac{l\sin\beta }{2}\right]+0.5k_{o}f_{ct}bh[h-(l\sin\beta )]}{a+0.5(l\sin\beta )\cot\beta }$$

### 4.2. Foster Model

The Foster model is a mechanical model utilized for the derivation of the shear strength of high-strength concrete corbels up to 105 MPa. This model is applicable for 30 high-performance concrete corbels with concrete strength range of 45 to 105 MPa, taking into account the secondary reinforcement. This model is implemented in this study for the prediction of the ultimate shear capacity of HyFRC, ECC, and SFRC corbels. The plastic truss model developed by Foster can be represented by three equations (Equations (4)–(6)).
(4)$$V_{u}=min\;(\frac{\rho_{s}f_{sy}w}{\Omega };\frac{f_{c}^{*}w}{d})$$
(5)$$\Omega =d-\sqrt{d^{2}-2aw-w^{2}}$$

For corbels with $\frac{a}{d}<2.0$:(6)fc*=min[[1.25−fc′500−0.72ad+0.18(ad)2]fc′;0.85fc^′]
where w = the bearing plate width, $f_{c}^{*}$ = the effective strength of the concrete compression strut, and Ω = the effective anchorage depth.

### 4.3. Comparison of Experimental and Analytical Results

The experimental results and the analytical models proposed by [7,12] were comparatively analyzed as shown in Table 8, Figure 15 and Figure 16. As observed from this table, the analytical results indicated a good consistency with the experimental results. The calculated and measured values of shear capacity are relatively closer, especially for HyFRC corbels. Moreover, the predicted shear capacity for all corbels using the truss model developed by Foster et al., showed a better a correlation with the measured results compared with the Fattuhi model. The latest model was derived from the Hagberg [31] model to consider the effects of steel fibers on the concrete strength by subjecting the corbels to tensional stress to obtain the indirect tensile splitting strength of the concrete samples. Thus, in this study, the developed Foster model shows the most satisfactory predictions for all HyFRC, ECC, and SFRC corbels, whereas according to literature review, Fattuhi model has been more consistent for high strength corbels and ultra-high performance concrete (UHPC) corbels [8].

The ultimate shear capacity of corbels was determined using the plastic truss method, which involves the use of a concrete efficiency factor model [12] for 30 high-strength concrete corbels. The compressive strengths of the corbels varied from 45 MPa to 105 MPa in the absence of steel fibers. The concrete efficiency factor model utilizes both shear span-to-effective depth ratio (a/d) and concrete compressive strength. Moreover, as indicated in Table 8, the predicted results using the formula are closer to the experimental results of the corbels with lower aspect ratio of 0.75, where shear is more dominant. Moreover, the predicted results slightly overestimate the shear strength of SFRC corbels, and slightly underestimates the strength of ECC corbels. However, there is a good agreement between the experimental and analytical results for HyFRC corbels.

As shown in this study, using the Foster method to determine the ultimate shear capacity of HyFRC, ECC, and SFRC corbels provides a reasonably low coefficient of variation of 13.8%, and modification on this model can be achieved in the future taking into consideration the effect of the intensive presence of steel and PVA fibers in concrete to get a more reliable model for prediction of the ultimate strength of HyFRC, ECC, and SFRC corbels.

## 5. Conclusions

In this study, the effect of using steel and PVA fibers in various self-compacting HPCC corbels including ECC, SFRC, and HyFRC were investigated. Failure modes, concrete crack patterns, displacement ductility with and without presence of closed stirrups or secondary reinforcements for different shear to depth (a/d) ratios were experimentally investigated and compared. Moreover, the validity of existing analytical models to predict the ultimate shear strengths of self-compacting HPCC corbels were investigated. From the experimental results, a beam-shear failure was observed in all high-performance cementitious composite (ECC, HyFRC, and SFRC corbels). The failure mode of the corbels is stable, as well as ductile. However, none of the corbels failed in a shear-friction mode of failure. Compared to the behaviour of conventional normal strength corbels, the high-performance composite corbels in this study showed insignificant loss in strength after the peak load.

Corbels with secondary reinforcement showed high shear strength capacity and slightly higher ductility compared to the corbels without stirrup. The provision of secondary reinforcement also decreased the crack width and improve the corbel stiffness. However, for the specimens with a/d of 1.0, where flexure is more dominant, the provision of secondary reinforcement causes more numerous cracks. Moreover, the shear span-to-depth ratio (a/d) also affects the shear behaviour of the corbels. In addition, increasing the a/d ratio of corbels increased the deflection, but decreased the shear capacity. The results indicated that the secondary reinforcement initiated a considerable difference in structural behaviour in terms of ultimate strength, ductility and failure, and formation of crack patterns between ECC, HyFRC, and SFRC corbels. In contrast, without secondary reinforcement, the corbels exhibited approximately similar stiffness and strength. Moreover, the results indicated that when the shear is more dominant, with the presence of secondary reinforcement, the ECC corbel shows superior performance than the other high-performance cementitious composite corbels. On the other hand, when the flexural is dominant (a/d = 1.0) and there is secondary reinforcement, SFRC corbels exhibited the lowest stiffness and ultimate strength compared to other cementitious composite corbels investigated in this study. In addition, HyFRC specimens showed slightly lower strength for the specimens with a/d = 0.75, where shear is higher. However, they exhibited higher deflection capacity and ductility for the specimens with higher a/d, where flexure is more dominant. Moreover, in terms of concrete cracks and damages, ECC and HyFRC corbels showed minor damage and fewer cracks induced compared to SFRC corbels.

In terms of ductility, ECC corbels for both a/d of 0.75 and 1.0 exhibited approximately a constant value between the range of 2 to 2.5, whereas SFRC corbels showed the greatest level of ductility, followed by HyFRC corbels.

Furthermore, the strength increase of all cementitious composite corbels was compared. It was shown that the SFRC corbel, among the other corbels, displayed the highest impact with 51% increment, followed by the ECC corbel with 45%, and lastly the HyFRC with 29% increment.

The experimental results demonstrated that fewer cracks and only minor damage were induced to the ECC and HyFRC corbels, Moreover, as displayed in the figure, compared to SFRC corbels, fewer and finer cracks were induced to the ECC and HyFRC corbels, which make them superior in terms of durability, service life, and sustainability compared to SFRC corbels. The comparison of the analytical models and the measured shear capacity of corbels shows very slight variation, which provides reasonable arguments for the suitability of the models of Fattuhi [7] and Foster et al. [12] for the design of corbels made of HyFRC, ECC, and SFRC. However, generalization of the models needs more research and empirical evidence.

## Figures and Tables

**Figure 1 materials-12-04159-f001:**
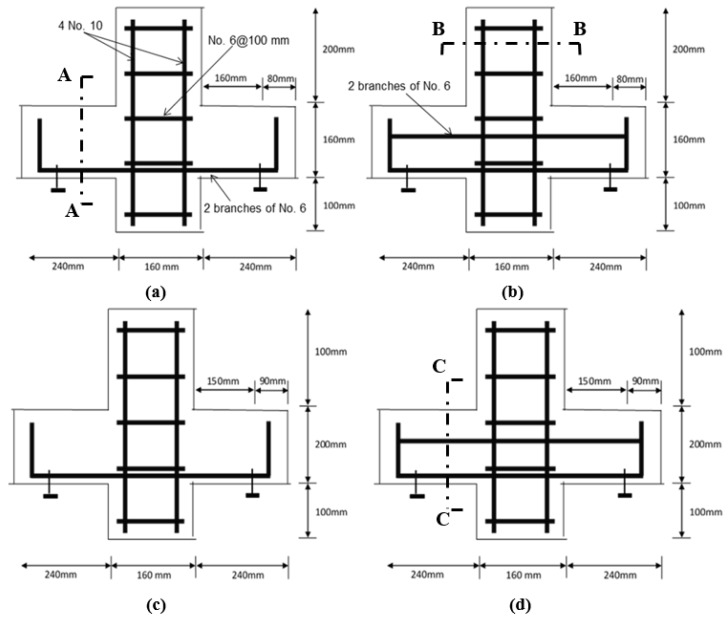
Corbel details (**a**) P 1.0, (**b**) S 1.0 (**c**) P 0.75, and (**d**) S 0.75.

**Figure 2 materials-12-04159-f002:**
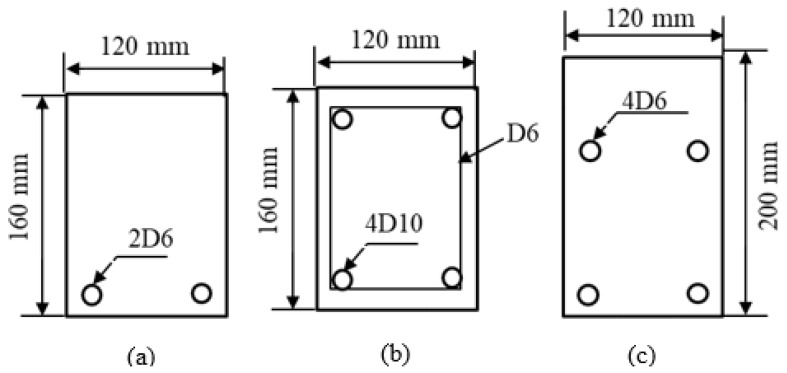
Cross sections shown in Figure 1. (**a**) Section A-A (corbels series “P”), (**b**) Section B-B (columns), and (**c**) Section C-C (corbels series “S”).

**Figure 3 materials-12-04159-f003:**
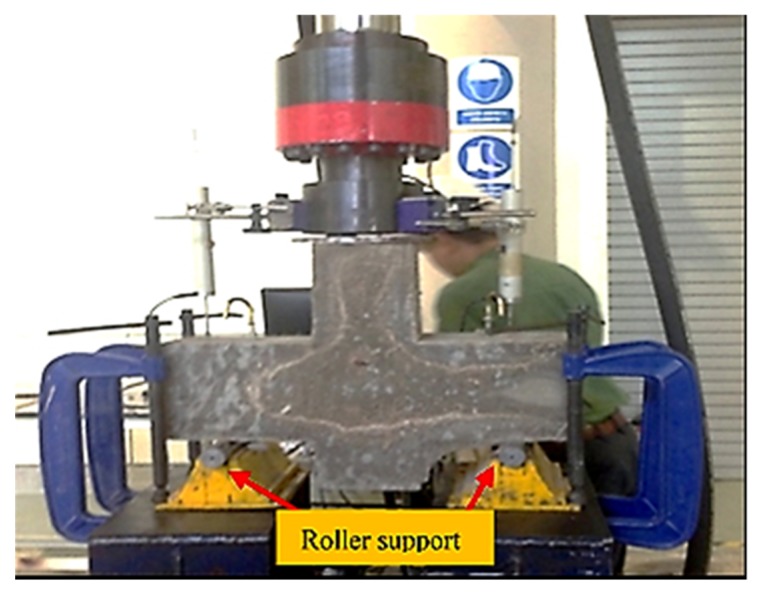
Setup for three-point bending test.

**Figure 4 materials-12-04159-f004:**
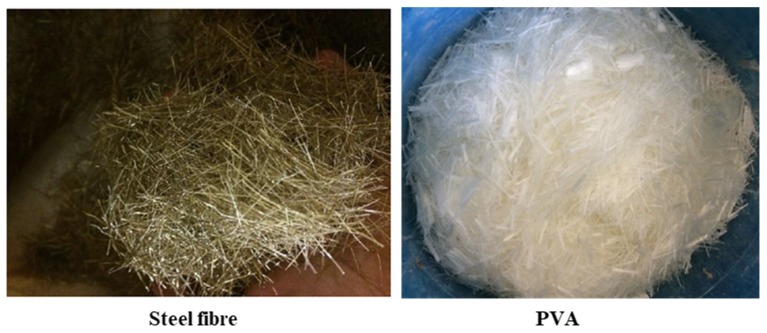
Steel fiber and PVA used in the test.

**Figure 5 materials-12-04159-f005:**
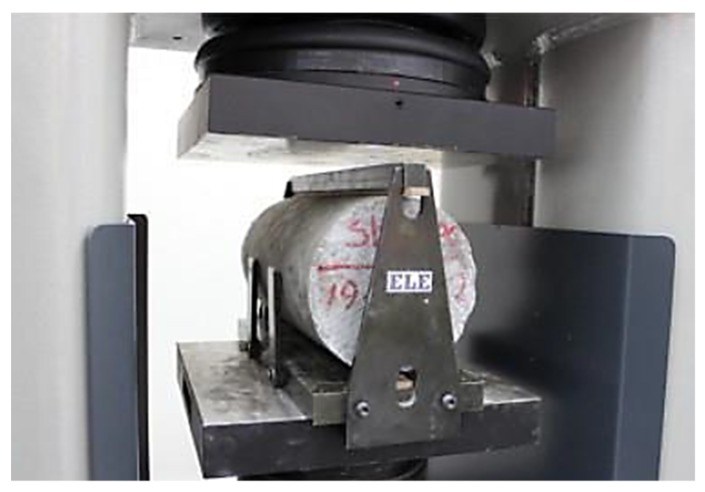
Setup of indirect tensile test.

**Figure 6 materials-12-04159-f006:**
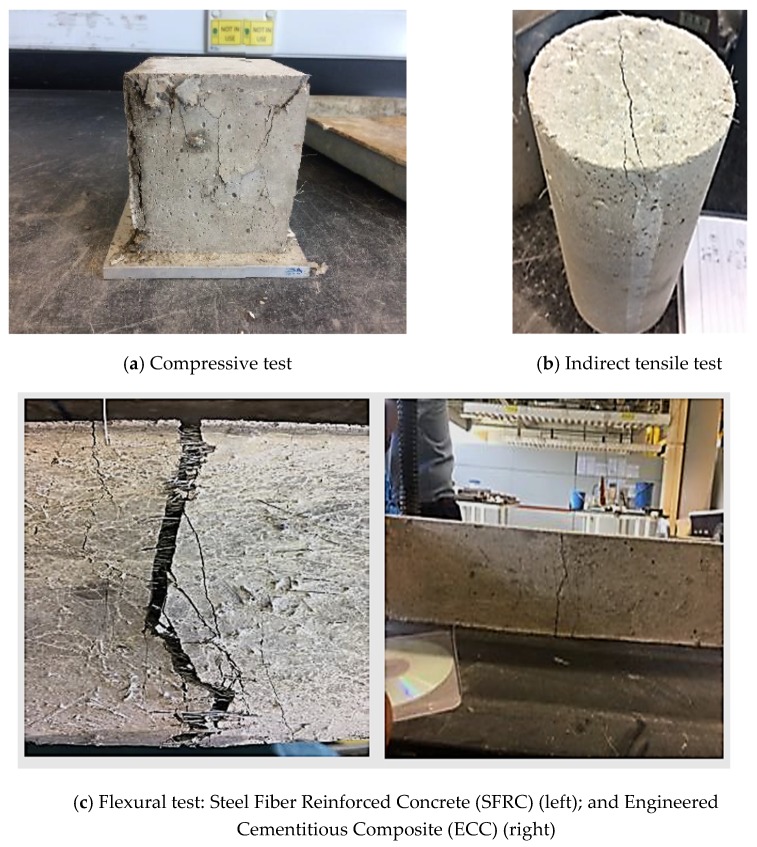
Sample of failures of specimens after (**a**) compressive, (**b**) indirect tensile, and (**c**) flexural tests.

**Figure 7 materials-12-04159-f007:**
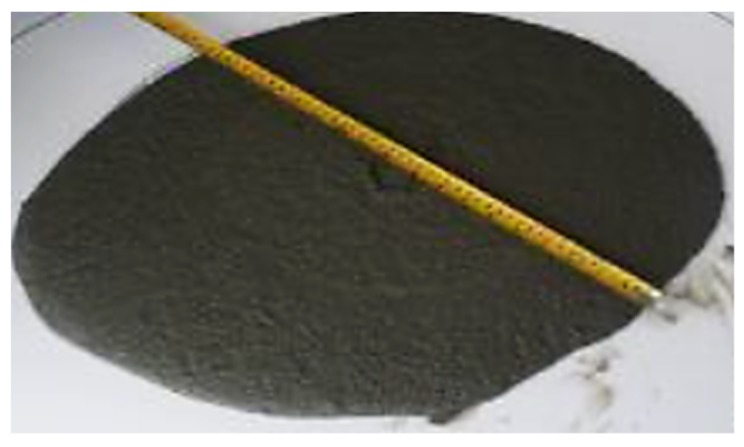
Sample of slump flow test conducted.

**Figure 8 materials-12-04159-f008:**
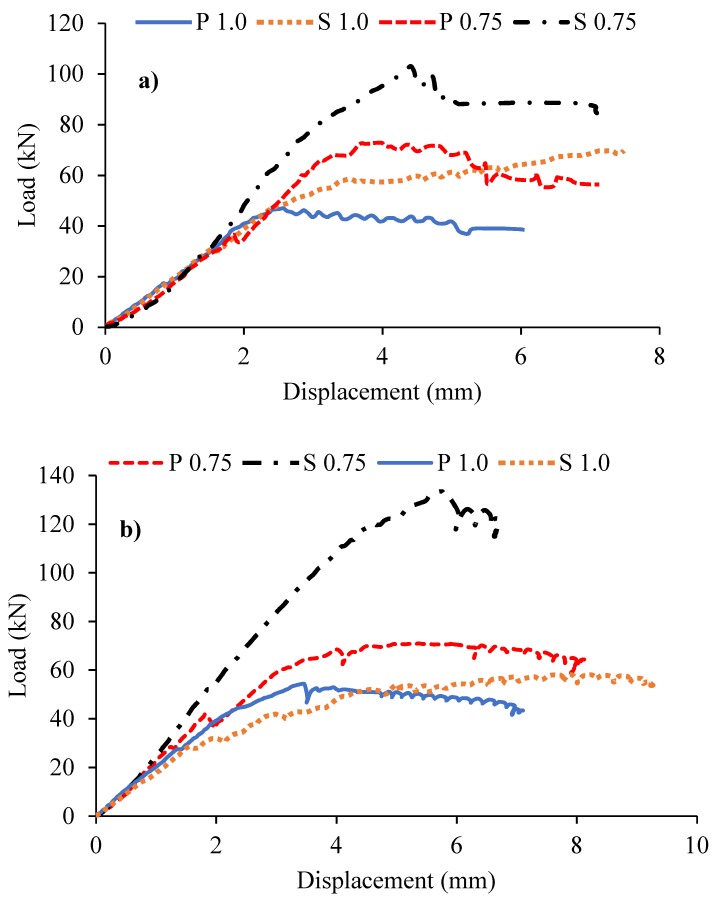
Influence of secondary reinforcement on (**a**) Hybrid Fiber Reinforced Concrete (HyFRC); (**b**) SFRC; and (**c**) ECC corbels.

**Figure 9 materials-12-04159-f009:**
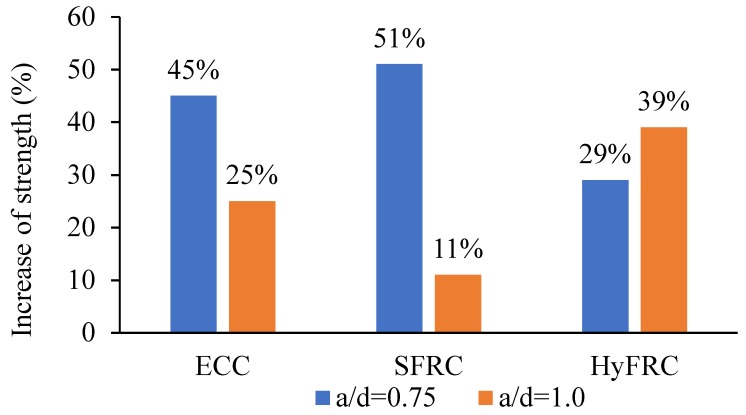
Strength increase of High Performance Fiber Reinforced Concrete (HPFRCC) corbels due to presence of secondary reinforcement.

**Figure 10 materials-12-04159-f010:**
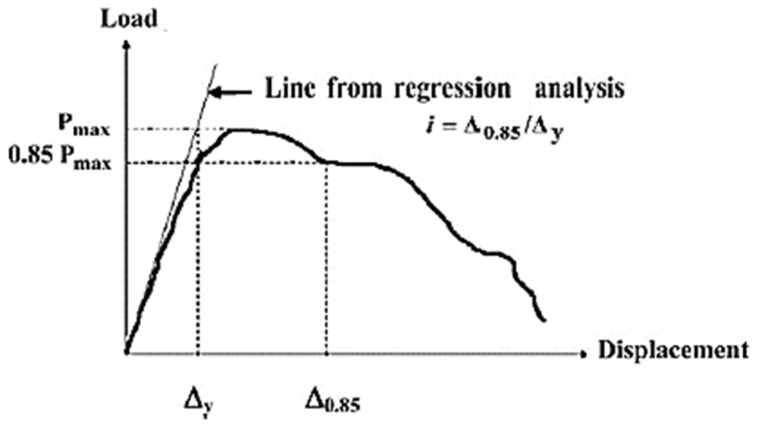
Displacement ductility for reinforced concrete structural elements [29,30].

**Figure 11 materials-12-04159-f011:**
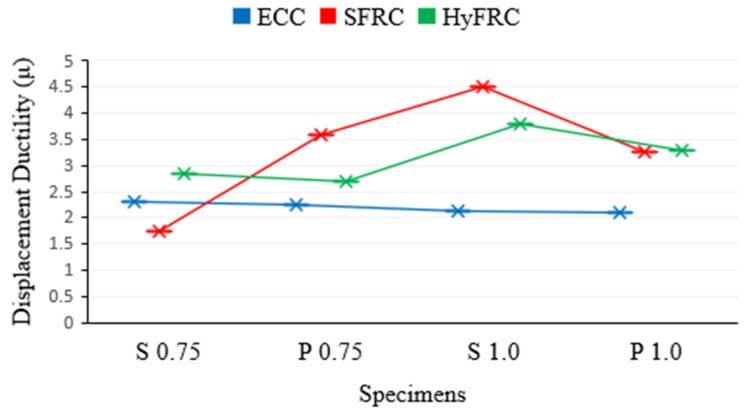
Displacement ductility of ECC; SFRC; and HyFRC corbels.

**Figure 12 materials-12-04159-f012:**
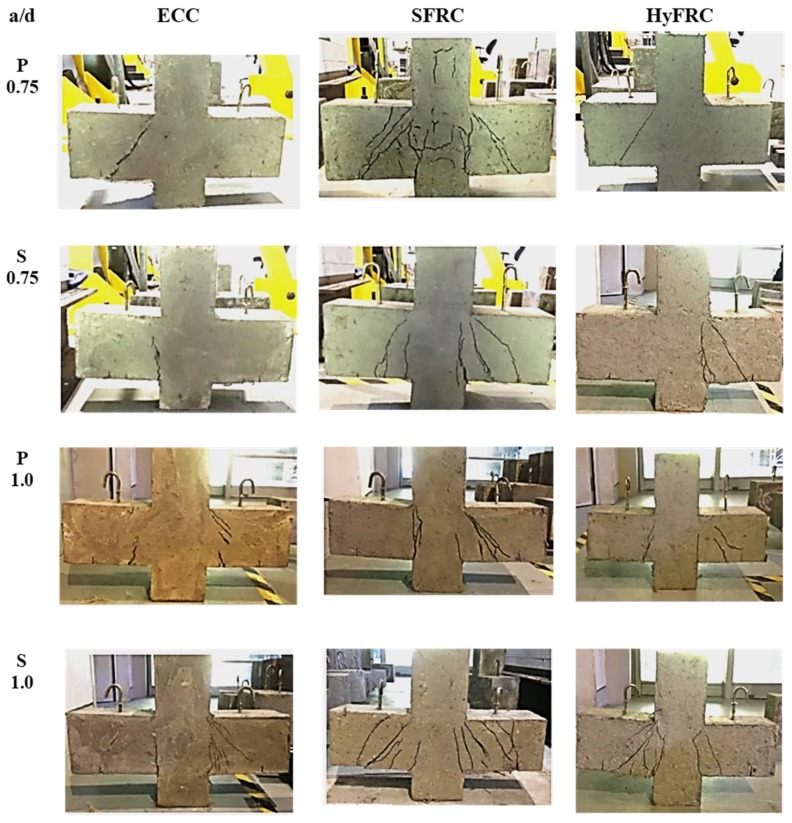
Cracks pattern of ECC; SFRC; and HyFRC corbels.

**Figure 13 materials-12-04159-f013:**
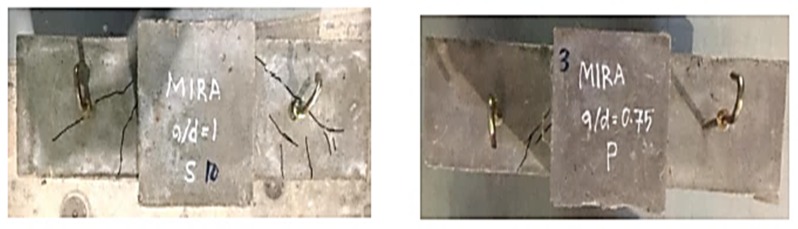
Cracks on the compression zones of SFRC: S 1.0 (left) and P 0.75 (right).

**Figure 14 materials-12-04159-f014:**
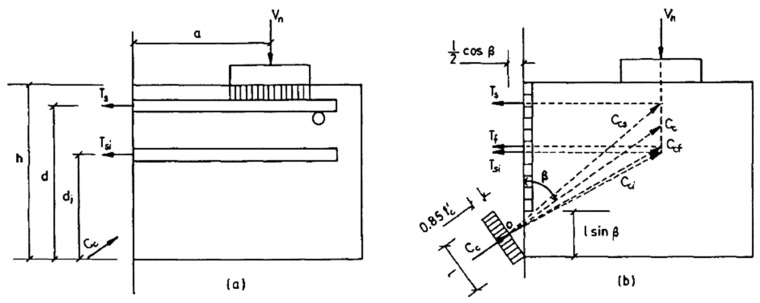
Truss model (**a**) stress distribution and (**b**) forces [22].

**Figure 15 materials-12-04159-f015:**
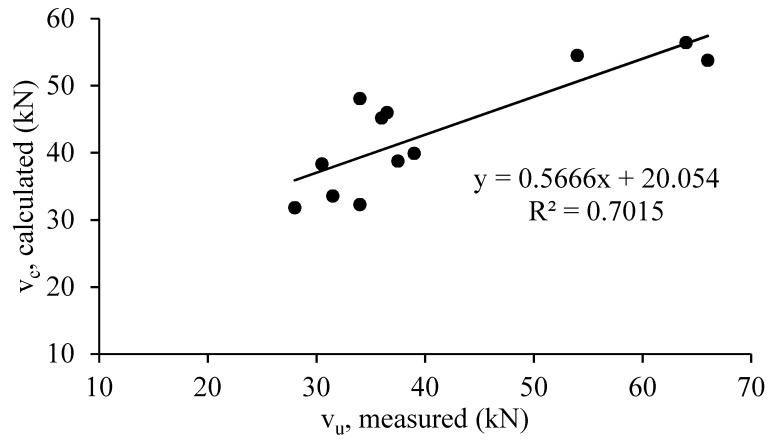
Comparison of Fattuhi [7] shear model with the experimental results.

**Figure 16 materials-12-04159-f016:**
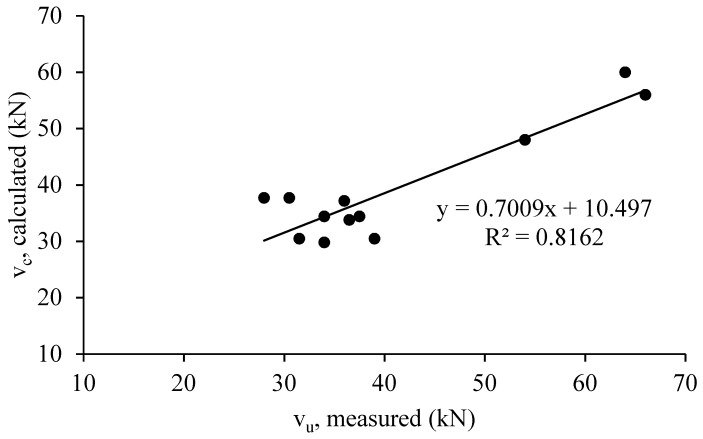
Comparison of Foster et al. [12] model with the experimental results.

**Table 1 materials-12-04159-t001:** Dimensions description of the specimens.

Corbel	Width (b), mm	Shear Span (a), mm	Depth (d), mm	a/d
P 1.0	120	160	160	1.00
P 0.75	120	150	200	0.75
S 1.0	120	160	160	1.00
S 0.75	120	150	200	0.75

**Table 2 materials-12-04159-t002:** Mixture design used in the experiments.

Ingredients (kg/m^3^)		HyFRC	ECC	SFRC
Binders	Type II, Ordinary Portland Cement	397 kg	547	377
	Fly Ash	131 kg	656	253
Aggregate	Coarse	414 kg	-	415
	Fine	1075 kg	438	793
Water		197 kg	312	253
Chemical Admixture (wt% binder)	Superplasticizers	0.6%	0.6%	0.6%
Fibers (vol%)	Steel	1.3%	-	1.6%
	PVA	0.3%	1.6%	-

**Table 3 materials-12-04159-t003:** Mechanical properties of the used reinforcing bars.

Nominal Diameter (mm)	Cross Sectional Area, as (mm^2^)	Yield Strength (MPa)	Ultimate Strength (MPa)
6	28.3	250	413
10	78.5	302	500

**Table 4 materials-12-04159-t004:** Properties of the fibers.

Fiber Type	Length (mm)	Diameter (mm)	Strength (MPa)
Steel	25	0.3	2300
Polyvinyl Alcohol (PVA)	12	0.04	1600

**Table 5 materials-12-04159-t005:** Flexural, tensile, compressive strength of fiber-reinforced cementitious composites. (Standard deviation value in parenthesis).

Age of Concrete→	7 Days			28 Days		
Type of Test↓	ECC	SFRC	HyFRC	ECC	SFRC	HyFRC
Flexural strength (MPa)	7.04 (0.55)	12.13 (0.04)	10.22 (0.55)	10.58 (0.44)	15.02 (0.17)	12.45 (1.04)
Indirect tensile strength (MPa)	4.74 (0.47)	4.94 (0.27)	5.11 (0.46)	5.54 (0.21)	5.59 (0.20)	5.41 (0.39)
Compressive strength (MPa)	46.75 (0.72)	54.37 (1.08)	48.00 (2.58)	49.70 (0.31)	65.17 (1.71)	57.58 (2.37)

**Table 6 materials-12-04159-t006:** Slump flow test results for the specimens.

Specimens	Slump Flow (mm)			Time 500 mm (S)
	Dmax	Dperp	Davg	
ECC	774	768	771	3.8
SFRC	698	688	693	5.1
HyFRC	724	718	721	4.2

**Table 7 materials-12-04159-t007:** Displacement ductility of the specimens (*i*).

Specimen	ECC			SFRC			HyFRC		
	$\Delta_{0.85}$	$\Delta_{y}$	*i*	$\Delta_{0.85}$	$\Delta_{y}$	*i*	$\Delta_{0.85}$	$\Delta_{y}$	*i*
S 0.75	7.8	3.4	2.3	6.4	3.8	1.7	5.9	2.1	2.8
P 0.75	6.3	2.9	2.2	8.2	2.3	3.6	5.9	2.2	2.7
S 1.0	6.9	3.3	2.1	9.6	2.1	4.5	7.6	2.0	3.8
P 1.0	5.6	2.7	2.1	8.1	2.5	3.2	5.4	1.6	3.3

**Table 8 materials-12-04159-t008:** Comparison of tested and calculated corbel results.

Specimen	Corbel Code	$V_{t}\;(\mathrm{kN})$	$V_{c}\;(\mathrm{kN})$			
			Fattuhi	$V_{t}/V_{c}$	Foster et al.	$V_{t}/V_{c}$
HyFRC	P 1.0	34.0	32.3	1.05	34.5	0.99
	P 0.75	36.5	46.0	0.79	33.8	1.08
	S 1.0	37.5	38.8	0.97	34.5	1.09
	S 0.75	54.0	54.5	0.99	48.0	1.13
ECC	P 1.0	31.5	33.6	0.94	30.5	1.03
	P 0.75	34.0	48.1	0.71	29.8	1.14
	S 1.0	39.0	39.9	0.98	30.5	1.28
	S 0.75	64.0	56.4	1.13	60.0	1.07
SFRC	P 1.0	28.0	31.8	0.88	37.7	0.74
	P 0.75	36.0	45.2	0.80	37.2	0.97
	S 1.0	30.5	38.4	0.80	37.7	0.81
	S 0.75	66.0	53.8	1.23	56.0	1.18

$V_{t}$ = experimental shear and $V_{c}$ = calculated shear.

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
