# Peer review of "Influence of Secondary Reinforcement on Behaviour of Corbels with Various Types of High-Performance Fiber-Reinforced Cementitious Composites"

_materials, 2019, doi:10.3390/ma12244159_

Round 1
Reviewer 1 Report
The reviewer judges that this manuscript does not reach the certain level for journal publications. Lots of lack of definitions, explanations, and considerations can be found out. The test results have been only compared with existing predicting method without explanations of details of fiber effect. Other comments are as follows:
Line 81: Please define “HPSCC”. Line 84: Please define “ECC”. Line 85: Please define “SRFC” and “HyHPC”. Figure 1: The figures and dimensions of the sections of the specimens should be described. Page 3: The measurement items and measurement methods must be described. Line 102: This line may be deleted. Page 4: The mechanical characteristics of used reinforcing bars must be explained. Page 4: The photos of used fibers should be added. Page 4: Fresh properties such as slump or flow test results must be described. Page 4: The details of compression, tension, and flexural test such as specimen dimensions, loading methods must be explained. Table 3: The definitions of flexural strength and tensile strength must be explained. Table 3: The numbers of the significant digits should be considered. Figure 4: The numbering order of the figures should be appearing order in body text. Figure 5: The reasons that the slope of the curves show slight increasing after 1 or 2 mm must be explained. Figure 7: The values for vertical axis must be added. Figure 7: The values of Delta-y and Delta-0.85 for all specimens must be listed in additional table. Lines 176, 178: “Figure 8” -> “Figure 7” ? Lines 176-177: The explained matter in the body text does not reflect Figure 7. Lines 194-196: Considerations the reason that only in the SFRC specimens shear cracks appeared in the top portion must be discussed. Was it by any irregularity of the loading, or ununiformity of the materials? Line 220: Explanation for “l” (first symbol in left term) must be added. Section 4.1 and 4.2 The methods taking the effect of fibers (HyFRC, ECC, SFRC) into account applying existing formulas must be clearly explained. Reference 1: Please check the name of Committee. Reference 19: Please check the name of journal.
Author Response
"Please see the attachment."

Reviewer 2 Report
Work may be considered for publication after the comments below are considered:
Section 2.2: How many test specimens were used for compressive and flexural strength testing? Table 1: Why was superplasticizer based on %cement weight instead of %cementitious? Table 2: Mpa -> MPa Please improve figure quality for Fig. 3b and 3c right. Table 3: What are the standard deviation values? Be consistent with the number of decimals. Why is Figure 5 referenced before Figure 4? The quality of Figure 4 is poor and it needs to be improved. The arrows in Figure 5 are confusing. I would just show the legend. Were any repeats tested? Quality of Figure 7, 8, 9 are poor. Show equation and R2 in Figure 11 and 12. Please comment on the implications of your work. Compare and contrast your findings with those from literature - the discussion is weak.
Author Response
"Please see the attachment."

Reviewer 3 Report
The research presented in the article are very interesting, but some comments and explanation to the research should be added:
Please specify the class of Portland cement Please define the testing of flexural and indirect tensile strength (e.g. conditions of testing, type and dimensions of samples and press machine) Please explain why you obtained so different values of compressive strength for HPCCs, especially difference between HyFRC and SFRC (the water to binder for HyFRC is lower than in case of SFRC, what is an added value of replacement of some part of steel fibres by PVA fibres?). Flexural and tensile properties are also reduced for HyFRC. Moreover, the characteristic of load – deflection for HyFRC presented in fig. 5 are the worst from all tested samples. Please comment the role of added randomly distributed reinforcement The quality of figures 4, 7 and 10 should be improve.
Author Response
"Please see the attachment."

Round 2
Reviewer 1 Report
The reviewer judges that this manuscript does not reach the certain level for journal publications. The test results have been only compared with existing predicting method without explanations of details of fiber effect. Same comments for first manuscript have been also raised up in the second manuscript as following comments (#3, #10, #11, #12, #13, #14, and #15). 1. Table 1: "Width" is mentioned as "breadth" in line 103. The reviewer thinks "Width" is better.
2. Table 1: What is the difference between "Height" and "Depth"?
3. Figure 1: The figures of the sections of the specimens should be added.
4. Line 115: "linear variable differential transformers" -> "linear variable displacement transducers" ?
5. Table 3: "Area" -> "Cross-sectional area"
6. Table 3: "Ultimate strength" -> "Tensile strength"
7. Line 141: Is ASTM D638-14 proper reference for the indirect tensile strength of concrete?
8. Table 6: Please add the unit for slump flow.
9. Line 168: "among" -> "Among"
10. Figure 7 and lines 194-199: The reviewer's comment for the first manuscript was about the reason that the slope of the curves show slight increasing after 1 or 2 mm comparing with that before 1mm in same specimens.
11. Table 7: The values of Delta-y and Delta-0.85 for all specimens must be listed.
12. Figure 9 and 10: The numbering order of the figures should be appearing order in the body text. The figure numbers in the body text should be also checked (Lines 223-225).
13. Line 225-226: The explained matter in the body text does not reflect Figure 10.
14. Line 282: Explanation for "l" (first symbol in left term) must be added. The reviewer does not think that [l sin beta] is one term. From Figure 13(b), it is considered that "l" indicates the dimension of compressive zone of concrete.
15. Reference 1: Please check the name of Committee.
Author Response
"Please see the attachment."

Reviewer 2 Report
I still think the quality of several of the figures could be significantly improved. In addition, results could still be better compared with literature and further insights provided.
